# Association between estimated plasma volume status and acute kidney injury in patients who underwent coronary revascularization: A retrospective cohort study from the MIMIC-IV database

**Xinping Yang, Fan Zhang, Yongqiang Zhan, Zhiheng Liu, Wenjing Wang, Jiahua Shi** [ID] *

Department of Anesthesiology, Shenzhen Second People's Hospital, The First Affiliated Hospital of Shenzhen University, Shenzhen, China

* sjiahdoc@outlook.com

**Data Availability Statement:** The datasets generated and analyzed during the current study

## Abstract

### Background

Acute kidney injury (AKI) remains a common complication of coronary revascularization and increases poor outcomes in critically ill surgical patients. Compared to the plasma volume status (PVS), estimated plasma volume status (ePVS) has the advantages of being noninvasive and simple and has been shown to be associated with worse prognosis in patients undergoing coronary revascularization. This study was to evaluate the association of ePVS with the risk of AKI in patients who underwent coronary revascularization.

### Methods

In this retrospective cohort study, data of patients who underwent coronary revascularization were extracted from the Medical Information Mart for Intensive Care (MIMIC)-IV database (2008–2019). The outcome was the occurrence of AKI after ICU admission. The covariates were screened via the LASSO regression method. Univariate and multivariate Logistic regression models were performed to assess the association of ePVS and PVS and the odds of AKI in patients who underwent coronary revascularization, with results shown as odds ratios (ORs) and 95% confidence intervals (CIs). Subgroup analyses of age, surgery, and anticoagulation agents and sequential organ failure assessment (SOFA) score were performed to further explore the association of ePVS with AKI.

### Results

A total of 3,961 patients who underwent coronary revascularization were included in this study, of whom 2,863 (72.28%) had AKI. The high ePVS was associated with the higher odds of AKI in patients who received coronary revascularization (OR = 1.06, 95%CI: 1.02–1.10), after adjusting for the covariates such as age, race, SAPS-II score, SOFA score, CCI, weight, heart rate, WBC, RDW-CV, PT, BUN, glucose, calcium, PH, PaO$_2$, mechanical ventilation, vasopressors, and diuretic. Similar results were found in patients who underwent

are available from the MIMIC-IV database, https://mimic.mit.edu/docs/iv/. The data are also available in the Supporting Information files.

**Funding:** The author(s) received no specific funding for this work.

**Competing interests:** The authors have declared that no competing interests exist.

the CABG (OR = 1.07, 95%CI: 1.02–1.11), without anticoagulation agents use (OR = 1.07, 95%CI: 1.03–1.12) and with high SOFA score (OR = 1.10, 95%CI: 1.04–1.17). No relationship was found between PVS and the odds of AKI in patients who underwent the coronary revascularization.

## Conclusion

The ePVS may be a promising parameter to evaluate the risk of AKI in patients undergoing coronary revascularization, which provides a certain reference for the risk stratification management of ICU patients who underwent coronary revascularization.

## Introduction

Coronary revascularization is the mainstay of treatment for coronary artery related disease, including percutaneous coronary intervention (PCI) and coronary artery bypass grafting (CABG) [1–3]. With the development and application of drug-eluting stents and minimally invasive surgery, the prognosis of patients undergoing PCI or CABG has improved, but some patients still suffer from postoperative complications, resulting in a poorer prognosis [4–7]. Acute kidney injury (AKI) is a common complication in patients undergoing coronary revascularization, with incidence ranging from 4% to 28% [8, 9]. AKI predicts a worse clinical prognosis and places a significant economic burden on patients and the healthcare system [10–12]. Identifying patients at risk for AKI is critical to the prognostic management and improvement of patients undergoing coronary revascularization.

The high load of plasma volume status (PVS) may contribute to the poor prognosis of coronary artery diseases through the over-activation of the renin-angiotensin-aldosterone system and could be used as a therapeutic target to improve their prognosis [13, 14]. However, PVS is difficult to quantify in a noninvasive way [15, 16]. Estimated plasma volume status (ePVS) is novel markers of congestion based on hemoglobin and hematocrit calculations with the characteristics of low-cost, rapidly quantified, which can monitor plasma volume fluctuations without total volume overload [17]. High ePVS was associated with congestion, cardiorenal syndrome, and prognosis. Furthermore, a decrease in ePVS was associated with a decrease in volume overload that is, decongestion, and with improved prognosis [18, 19]. Previous study has reported that high ePVS was significantly associated with an increased risk of acute and chronic renal failure in acute myocardial infarction (AMI) patients after CABG [20]. High ePVS was associated with an increased risk of contrast-associated nephropathy (CIN) in heart failure (HF) patients undergoing PCI [19]. To the best of our knowledge, however, the relationship between ePVS and the risk of AKI in patients who underwent coronary revascularization (CABG and/or PCI) remains unclear.

This study aimed to investigate the association between the ePVS and the risk of AKI in patients who underwent coronary revascularization and whether this association remains in patients stratified by age, surgery, anticoagulation agents and sequential organ failure assessment (SOFA) score, which may provide certain reference for the risk stratification management in patients who underwent coronary revascularization.

## Methods

### Study design and participants

Data of patients who underwent coronary revascularization in this retrospective cohort study was extracted from the Medical Information Mart for Intensive Care (MIMIC)-IV database

(https://mimic.mit.edu/docs/iv/). The database is a single-center and open-access database that covers deidentified related data of patients admitted to the intensive care unit (ICU) the Tertiary Academic Medical Center in Boston, MA, USA from 2008–2019. The database was approved by the Institutional Review Boards of Beth Israel Deaconess Medical Center (Boston, MA) and the Massachusetts Institute of Technology (Cambridge, MA). To protect patient privacy, all private information in the database depository was anonymized. The requirement of ethical approval for this was waived by the Institutional Review Board of Shenzhen Second People's hospital, The First Affiliated Hospital of Shenzhen University, because the data was accessed from MIMIC-IV (a publicly available database). The need for written informed consent was waived by the Institutional Review Board of Shenzhen Second People's hospital, The First Affiliated Hospital of Shenzhen University due to retrospective nature of the study. All methods were performed in accordance with the relevant guidelines and regulations.

The inclusion criteria were: (1) aged ≥18 years old; (2) ICU patients who received coronary revascularization. The exclusion criteria were: (1) patients with ICU stay less than 24 hours; (2) patients with missing data of hematocrit and hemoglobin upon ICU admission; (3) patients with missing AKI assessment after surgery during ICU stay; (4) patients diagnosed as AKI before ICU admission; (5) patients diagnosed with end-stage renal disease (ESRD); (6) patients missing body weight data; (7) patients with polycythemia upon ICU admission.

## PVS and ePVS definition

PVS was defined as follows: actual plasma volume (APV) = (1-hematocrit) × [a + (b × body weight)] (a = 1,530 in males and a = 864 in females, b = 41.0 in males and b = 47.9 in females). Ideal plasma volume (iPV) = c × body weight (c = 39 in males and c = 40 in females). PVS = [(APV-iPV)/iPV × 100(%)].

Hematocrit, and hemoglobin were measured within 24 hours on admission to ICU. ePVS was calculated from the Strauss-derived Duarte formula [17]: ePVS = [100-hematocrit (%)]/hemoglobin (g/dL).

## Potential covariates

The potential covariates was extracted as follows: age, gender (female/male), race (Asian, Black, Hispanic/Latino, White, unknown or others), insurance status (medicaid/medicare or others), marital status (married, single/divorced/widowed, or unknown), 24-h urine output, simplified acute physiology score II (SAPS-II), sequential organ failure assessment score (SOFA) score, glasgow coma scale (GCS) score, charlson comorbidity index (CCI), weight, heart rate, systolic blood pressure (SBP), diastolic blood pressure (DBP), respiratory rate, temperature, white blood cells (WBC), platelet, red blood cell distribution width-coefficient of variation (RDW-CV), creatinine, estimated glomer ular filtration rate (eGFR), international normalized ratio (INR), prothrombin time (PT), partial thromboplastin time (PTT), blood urea nitrogen (BUN), glucose, calcium, sodium, potassium, chloride, bicarbonate, anion gap, lactate, pondus hydrogenii (PH), pressure of alveolar carbon dioxide ($PaCO_2$), pressure of alveolar oxygen ($PaO_2$), mechanical ventilation, vasopressors, thrombolysis, anticoagulation agents, diuretic, and surgery (PCI, CABG, or PCI/CABG).

## Outcome and follow-up

AKI was defined according to the Kidney Disease Improving Global Outcomes (KDIGO) [21] as follows: (1) an increase in serum creatinine (SCr) level ≥0.3 mg/dl within 48 hours; (2) an increase in SCr level to ≥1.5 times than the level at ICU admission within 7 days; (3) urine volume < 0.5mL/kg/h for 6 hours.

The outcome was the occurrence of AKI after ICU admission. The follow-up endpoint was the occurrence of AKI during ICU stay or ICU discharge. The median follow-up time was 0.86 (0.60, 1.25) days.

## Statistical analysis

Continuous variables with normal distribution were described as mean ± standard deviation (SD), and between-group differences were compared by Student's *t* test or Satterthwaite *t* test. Data are presented as medians and quartiles [M (Q1, Q3)] for continuous variables with skewed distribution, and compared using Mann–Whitney U-test. Categorical variables were represented as number and percentage [n (%)], and Chi-square and Fisher's exact test were used for comparison between the groups.

The missing values were carried out using the interpolation. Sensitivity analyses were performed on the missing data before and after interpolation (S1 Table). The covariates were screened via the LASSO regression method, including age, race, SAPS-II, SOFA score, CCI, weight, heart rate, WBC, RDW-CV, PT, BUN, glucose, calcium, PH, $PaO_2$, mechanical ventilation, vasopressors, and diuretics. The result of collinearity tests was shown in S2 Table. The screening process is shown in S1 Fig. Univariate and multivariate Logistic regression models were used to assess the association between ePVS and PVS and the risk of AKI in patients who underwent coronary revascularization, with odds ratios (ORs) with 95% confidence intervals (CIs). The associations were further explored in different subgroups of age, surgery, and anticoagulation agents. All statistical analyses were performed using SAS 9.4 (SAS Institute Inc., Cary, NC, USA) and R version 4.2.2 (Institute for Statistics and Mathematics, Vienna, Austria). Statistical differences were considered when the *P*-value was <0.05.

## Results

### Characteristics of patients who underwent coronary revascularization

The flowchart of study participants was presented in Fig 1. We excluded 629 patients with ICU stay less than 24 hours, 19 patients with missing data of hematocrit and hemoglobin, 1,863

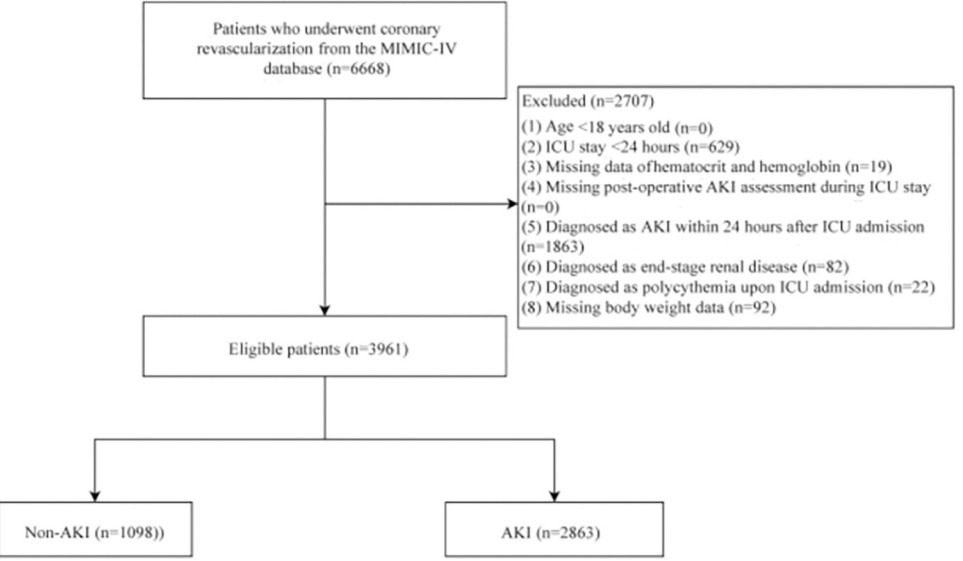

**Fig 1. The screening flowchart of patients who underwent coronary revascularization.**

patients diagnosed with AKI within 24 hours after ICU admission, 82 patients diagnosed with ESRD, 22 patients diagnosed as polycythemia upon ICU admission and 92 patients missing body weight data. Then 3,961 eligible patients were finally included, of which 2,863 (72.28%) patients suffered from AKI. Table 1 summarizes the characteristics of patients stratified according to whether AKI occurred or not. The mean age of all patients who underwent coronary revascularization was 67.68 ± 10.87 years old, with the median follow-up time was 0.86 (0.60, 1.25) days. The ePVS level in without AKI group was lower than in AKI group (6.74 ± 2.21 *vs.* 7.38 ± 2.42) ($P$ <0.001). There was no statistically difference in PVS level between two groups ($P$ >0.05). There were significant differences between the two groups with respect to age, race, insurance status, 24-h urine output, blood diseases, anemia, SAPS-II score, SOFA score, CCI score, weight, heart rate, SBP, DBP, temperature, WBC, platelet, hematocrit, hemoglobin, RDW-CV, creatinine, eGFR, INR, PT, BUN, glucose, calcium, sodium, potassium, chloride, bicarbonate, PH, $PaCO_2$, $PaO_2$, mechanical ventilation, vasopressors use, thrombolysis, diuretics use and surgery (all $P$ < 0.05). The comorbidities of CCI in AKI group and non-AKI group were shown in S3 Table.

## Association between ePVS and the odds of AKI in patients who underwent coronary revascularization

The relationship between ePVS and the odds of AKI in patients who underwent coronary revascularization was presented in Table 2. When each unit of ePVS was increased, the risk of AKI in patients received coronary revascularization increased by 0.06 times (OR = 1.06, $P$ = 0.008), after adjustments for age, race, SAPS-II score, SOFA score, CCI, weight, heart rate, WBC, RDW-CV, PT, BUN, glucose, calcium, PH, $PaO_2$, mechanical ventilation, vasopressors, and diuretics. It was indicated that high ePVS was associated with the higher odds of AKI in patients who underwent coronary revascularization.

## Association between PVS and the odds of AKI in patients who underwent coronary revascularization

Table 3 depicts the relationship between PVS and the risk of AKI in patients who underwent the coronary revascularization. We observed that there was no statistically difference between PVS and the odds of AKI in patients who underwent coronary revascularization after adjustment for age, race, blood disease, SAPSII, SOFA, CCI, RDW-CV, eGFR, INR, glucose, calcium, pH, PaCO2, mechanical ventilation, vasopressors and diuretic ($P$ >0.05). The receiver operator characteristic (ROC) of ePVS and PVS was presented in Fig 2. Taken together, monitoring the ePVS level and the odds of AKI may be more clinically meaningful than monitoring the PVS level among patients who underwent the coronary revascularization.

## Association between ePVS and the odds of AKI in patients who underwent coronary revascularization with different subgroups of age, surgery, and anticoagulation agents

This association was further explored in different subgroups of age, surgery, and anticoagulation agents. In Fig 3, the results showed that the high ePVS was associated with higher odds of AKI in patients who underwent the CABG (OR = 1.07, 95%CI: 1.02–1.11, P = 0.005), without anticoagulation agents use (OR = 1.07, 95%CI: 1.03–1.12, P = 0.010) and high SOFA score (OR = 1.01, 95%CI: 1.04–1.17). The ePVS level in different subgroups were shown in S4 Table.

**Table 1. Characteristics of patients who underwent coronary revascularization.**

| Variables | Total (N = 3961) | Non-AKI (N = 1098) | AKI (N = 2863) | P |
|---|---|---|---|---|
| Age, years, Mean ± SD | 67.68 ± 10.87 | 65.55 ± 11.02 | 68.49 ± 10.70 | <0.001 |
| Gender, n (%) | | | | 0.176 |
| Female | 959 (24.21) | 249 (22.68) | 710 (24.80) | |
| Male | 3002 (75.79) | 849 (77.32) | 2153 (75.20) | |
| Race, n (%) | | | | <0.001 |
| White | 2792 (70.49) | 720 (65.57) | 2072 (72.37) | |
| Asian | 85 (2.15) | 41 (3.73) | 44 (1.54) | |
| Black | 159 (4.01) | 52 (4.74) | 107 (3.74) | |
| Hispanic/Latino | 112 (2.83) | 49 (4.46) | 63 (2.20) | |
| Others | 152 (3.84) | 58 (5.28) | 94 (3.28) | |
| Unknown | 661 (16.68) | 178 (16.22) | 483 (16.87) | |
| Insurance status, n (%) | | | | <0.001 |
| Medicaid | 148 (3.74) | 61 (5.56) | 87 (3.04) | |
| Medicare | 1768 (44.64) | 441 (40.16) | 1327 (46.35) | |
| Other | 2045 (51.62) | 596 (54.28) | 1449 (50.61) | |
| Marital status, n (%) | | | | 0.447 |
| Married | 2396 (60.49) | 674 (61.38) | 1722 (60.15) | |
| Single/Divorced/Widowed | 1284 (32.42) | 355 (32.33) | 929 (32.45) | |
| Unknown | 281 (7.09) | 69 (6.29) | 212 (7.40) | |
| 24 h urine output, mL/d, M (Q$_1$, Q$_3$) | 1845.00 (1359.00–2475.00) | 2375.00 (1810.00–2989.25) | 1675.00 (1246.00–2195.00) | <0.001 |
| Blood diseases, n (%) | | | | <0.001 |
| No | 1878 (47.41) | 633 (57.65) | 1245 (43.49) | |
| Yes | 2083 (52.59) | 465 (42.35) | 1618 (56.51) | |
| Anemia, n (%) | | | | <0.001 |
| No | 634 (16.01) | 244 (22.22) | 390 (13.62) | |
| Yes | 3327 (83.99) | 854 (77.78) | 2473 (86.38) | |
| SAPS II, score, Mean ± SD | 35.68 ± 11.55 | 32.05 ± 11.08 | 37.07 ± 11.42 | <0.001 |
| SOFA, score, Mean ± SD | 5.15 ± 2.85 | 4.03 ± 2.46 | 5.58 ± 2.88 | <0.001 |
| GCS, score, Mean ± SD | 13.44 ± 3.46 | 13.55 ± 3.37 | 13.40 ± 3.50 | 0.224 |
| CCI, score, Mean ± SD | 2.19 ± 1.86 | 1.92 ± 1.75 | 2.29 ± 1.89 | <0.001 |
| Weight, kg, Mean ± SD | 85.72 ± 19.29 | 80.19 ± 16.25 | 87.84 ± 19.94 | <0.001 |
| Heart rate, bpm, Mean ± SD | 80.62 ± 11.30 | 79.70 ± 11.58 | 80.97 ± 11.17 | 0.002 |
| SBP, mmHg, Mean ± SD | 114.17 ± 18.39 | 116.24 ± 18.68 | 113.38 ± 18.22 | <0.001 |
| DBP, mmHg, Mean ± SD | 60.43 ± 12.42 | 63.17 ± 13.10 | 59.38 ± 11.98 | <0.001 |
| Respiratory rate, insp/min, M (Q$_1$, Q$_3$) | 15.00 (14.00–17.00) | 15.00 (14.00–18.00) | 15.00 (14.00–17.00) | 0.380 |
| Temperature, ˚C, Mean ± SD | 36.32 ± 0.62 | 36.37 ± 0.57 | 36.30 ± 0.63 | 0.002 |
| WBC, K/uL, M (Q$_1$, Q$_3$) | 12.30 (9.30–15.70) | 11.30 (8.70–14.70) | 12.60 (9.60–16.20) | <0.001 |
| Platelet, K/uL, Mean ± SD | 163.70 ± 63.38 | 171.94 ± 65.56 | 160.54 ± 62.24 | <0.001 |
| RDW-CV, %, Mean ± SD | 13.76 ± 1.36 | 13.59 ± 1.28 | 13.83 ± 1.39 | <0.001 |
| Creatinine, mg/dL, Mean ± SD | 0.94 ± 0.34 | 0.89 ± 0.29 | 0.96 ± 0.35 | <0.001 |
| eGFR, mL/min/1.73 m$^2$, Mean ± SD | 72.48 ± 23.21 | 76.04 ± 22.58 | 71.12 ± 23.31 | <0.001 |
| INR, M (Q$_1$, Q$_3$) | 1.40 (1.20–1.50) | 1.30 (1.20–1.50) | 1.40 (1.30–1.50) | <0.001 |
| PT, sec, M (Q$_1$, Q$_3$) | 15.10 (13.80–16.70) | 14.60 (13.20–16.10) | 15.30 (14.00–16.90) | <0.001 |
| PTT, sec, M (Q$_1$, Q$_3$) | 30.40 (27.30–35.30) | 30.30 (27.00–35.00) | 30.50 (27.40–35.50) | 0.131 |
| BUN, mg/dL, M (Q$_1$, Q$_3$) | 16.00 (13.00–20.00) | 15.00 (12.00–19.00) | 16.00 (13.00–21.00) | <0.001 |
| Glucose, mg/dL, M (Q$_1$, Q$_3$) | 135.00 (114.00–164.00) | 131.00 (109.25–156.00) | 137.00 (116.00–166.00) | <0.001 |
| Calcium, mmol/L, Mean ± SD | 2.38 ± 2.74 | 3.37 ± 3.43 | 2.00 ± 2.32 | <0.001 |

*(Continued)*

**Table 1.** (Continued)

| Variables | Total (N = 3961) | Non-AKI (N = 1098) | AKI (N = 2863) | P |
|---|---|---|---|---|
| Sodium, mEq/L, Mean ± SD | 135.62 ± 3.06 | 135.86 ± 3.07 | 135.53 ± 3.05 | 0.003 |
| Potassium, mEq/L, Mean ± SD | 4.61 ± 0.76 | 4.55 ± 0.76 | 4.64 ± 0.76 | 0.001 |
| Chloride, mEq/L, M (Q_1, Q_3) | 106.00 (104.00–108.00) | 105.00 (103.00–108.00) | 106.00 (104.00–108.00) | <0.001 |
| Bicarbonate, mEq/L, Mean ± SD | 23.13 ± 2.52 | 23.48 ± 2.40 | 22.99 ± 2.55 | <0.001 |
| Anion gap, mEq/L, Mean ± SD | 11.70 ± 3.08 | 11.71 ± 3.01 | 11.70 ± 3.10 | 0.969 |
| Lactate, mmol/L, M (Q_1, Q_3) | 2.10 (1.60–2.70) | 2.10 (1.60–2.70) | 2.10 (1.60–2.80) | 0.285 |
| PH, units, Mean ± SD | 7.40 ± 0.07 | 7.41 ± 0.06 | 7.40 ± 0.07 | <0.001 |
| PaCO_2, mmHg, Mean ± SD | 40.99 ± 6.34 | 40.47 ± 5.93 | 41.19 ± 6.48 | 0.001 |
| PaO_2, mmHg, Mean ± SD | 306.06 ± 105.43 | 315.24 ± 103.50 | 302.54 ± 105.97 | 0.001 |
| Mechanical ventilation, n (%) | | | | <0.001 |
|   No | 228 (5.76) | 134 (12.20) | 94 (3.28) | |
|   Yes | 3733 (94.24) | 964 (87.80) | 2769 (96.72) | |
| Vasopressors, n (%) | | | | <0.001 |
|   No | 1150 (29.03) | 472 (42.99) | 678 (23.68) | |
|   Yes | 2811 (70.97) | 626 (57.01) | 2185 (76.32) | |
| Thrombolysis, n (%) | | | | 0.264 |
|   No | 3886 (98.11) | 1082 (98.54) | 2804 (97.94) | |
|   Yes | 75 (1.89) | 16 (1.46) | 59 (2.06) | |
| Anticoagulation agents, n (%) | | | | <0.001 |
|   No | 3441 (86.87) | 862 (78.51) | 2579 (90.08) | |
|   Yes | 520 (13.13) | 236 (21.49) | 284 (9.92) | |
| Diuretic, n (%) | | | | <0.001 |
|   No | 1949 (49.20) | 636 (57.92) | 1313 (45.86) | |
|   Yes | 2012 (50.80) | 462 (42.08) | 1550 (54.14) | |
| Iron, n (%) | | | | 0.712 |
|   No | 3893 (98.28) | 1081 (98.45) | 2812 (98.22) | |
|   Yes | 68 (1.72) | 17 (1.55) | 51 (1.78) | |
| Surgery, n (%) | | | | <0.001 |
|   PCI | 573 (14.47) | 273 (24.86) | 300 (10.48) | |
|   CABG | 3367 (85.00) | 821 (74.77) | 2546 (88.93) | |
|   PCI & CABG | 21 (0.53) | 4 (0.37) | 17 (0.59) | |
| Hemoglobin, g/dL, Mean ± SD | 10.31 ± 2.25 | 10.77 ± 2.26 | 10.13 ± 2.22 | <0.001 |
| Hematocrit, %, Mean ± SD | 30.89 ± 6.65 | 32.24 ± 6.64 | 30.37 ± 6.59 | <0.001 |
| ePVS, Mean ± SD | 7.20 ± 2.38 | 6.74 ± 2.21 | 7.38 ± 2.42 | <0.001 |
| PVS, Mean ± SD | 0.04 ± 0.13 | 0.04 ± 0.14 | 0.04 ± 0.12 | 0.783 |

t: t-test; t': Satterthwaite t-test; W: Wilcoxon rank sum test; $\chi^2$: Chi-square test; -: Fisher exact; SD: standard deviation; M: Median, Q1:1st Quartile, Q3:3st Quartile. AKI: acute kidney injury; AMI: acute myocardial infarction; HF: heart failure; SAPS: simplified acute physiology score; SOFA: sequential organ failure assessment; GCS: glasgow coma scale; CCI: charlson comorbidity index; SBP: systolic blood pressure; DBP: diastolic blood pressure; WBC: white blood cell count; RDW-CV: red blood cell distribution width-coefficient of variation; eGFR: estimated glomerular filtration rate; INR: international normalized ratio; PT: prothrombin time; PTT: partial thromboplastin time; BUN: blood urea nitrogen; PH: pondus hydrogenii; PaCO2: pressure of alveolar carbon dioxide; PaO2: pressure of alveolar oxygen; ePVS: estimated plasma volume status; PCI: percutaneous coronary intervention; CABG: coronary artery bypass grafting.

## Discussion

The current study evaluated the association of ePVS with the risk of AKI in patients underwent coronary revascularization. Our findings showed that the high ePVS was associated with the higher odds of AKI in patients who underwent coronary revascularization. We also discovered

**Table 2. Association of ePVS with the odds of AKI in patients who underwent coronary revascularization.**

| Variables | Model 1 | | Model 2 | |
|---|---|---|---|---|
| | OR (95%CI) | P | OR (95%CI) | P |
| ePVS | 1.13 (1.10–1.17) | <0.001 | 1.06 (1.02–1.10) | 0.006 |

ePVS: estimated plasma volume status; AKI: acute kidney injury; OR: odds ratio; CI: confidence interval.

Model 1: univariate logistic regression model

Model 2: adjusted for age, race, SAPS-II score, SOFA score, CCI, weight, heart rate, WBC, RDW-CV, PT, BUN, glucose, calcium, PH, $PaO_2$, mechanical ventilation, vasopressors, and diuretics.

high ePVS was associated with higher odds of AKI in patients aged <65 years, undergoing CABG, and who did not use anticoagulation agents. We also observed that there was no relationship between PVS and AKI in patients who underwent coronary revascularization. It was suggested that the ePVS may be a promising parameter for AKI risk management in patients undergoing coronary revascularization.

ePVS is a marker of intravascular congestion that is indirectly related to plasma volume shifts at the interstitial tissue level [17, 22]. Previous researches have indicated that ePVS was a potential prognostic indicator associated with rehospitalization, and mortality in patients with AMI and HF [23–30]. Chen et al. [24] reported that higher ePVS value was associated with a higher risk of in-hospital mortality in acute AMI patients. Soetjoadi et al. [27] found that high ePVS was associated with in-hospital mortality in right HF. Fudim et al. [28] found that high ePVS was associated with in-hospital outcomes of decompensated HF. Recently, several studies have investigated ePVS in patients with AMI or HF undergoing PCI or CABG [19, 20]. He et al. [19] reported that the high ePVS was also associated with an increased risk of CIN and higher long-term mortality in HF patients undergoing PCI. Maznyczka et al. [20] found that higher preoperative ePVS was associated with an increased risk of acute and chronic renal failure in AMI patients undergoing CABG. However, the relationship between ePVS and the risk of AKI in patients who underwent coronary revascularization (CABG and/or PCI) remains unclear. The present study found that the high ePVS was associated with the higher odds of AKI in spatients who underwent coronary revascularization.

In addition, we also found the high ePVS was associated with higher odds of AKI in patients who underwent CABG, without anticoagulation agents use and with high SOFA score. The ePVS consists of erythrocyte pressure volume and hemoglobin, and the use of anticoagulants may prevent blood coagulation, affecting erythrocyte pressure volume and hemoglobin levels, and making the association between ePVS and AKI risk not statistically different. The role of ePVS in the risk of AKI among patients on anticoagulants needs to be further explored. Evidence suggests that administration of ipragliflozin resulted in reduced ePVS levels in patients with type 2 diabetes mellitus [31]. The high ePVS seemed to not be associated with a risk of

**Table 3. Association of PVS with the odds of AKI in patients who underwent coronary revascularization.**

| Variables | Model 1 | | Model 2 | |
|---|---|---|---|---|
| | OR (95%CI) | P | OR (95%CI) | P |
| PVS | 1.08 (0.62–1.87) | 0.783 | 1.84 (0.76–4.47) | 0.175 |

PVS: plasma volume status; AKI: acute kidney injury; OR: odds ratio; CI: confidence interval.

Model 1: univariate logistic regression model

Model 2: adjusted for age, race, blood disease, SAPSII, SOFA, CCI, RDW-CV, eGFR, INR, glucose, calcium, pH, PaCO2, mechanical ventilation, vasopressors and diuretic.

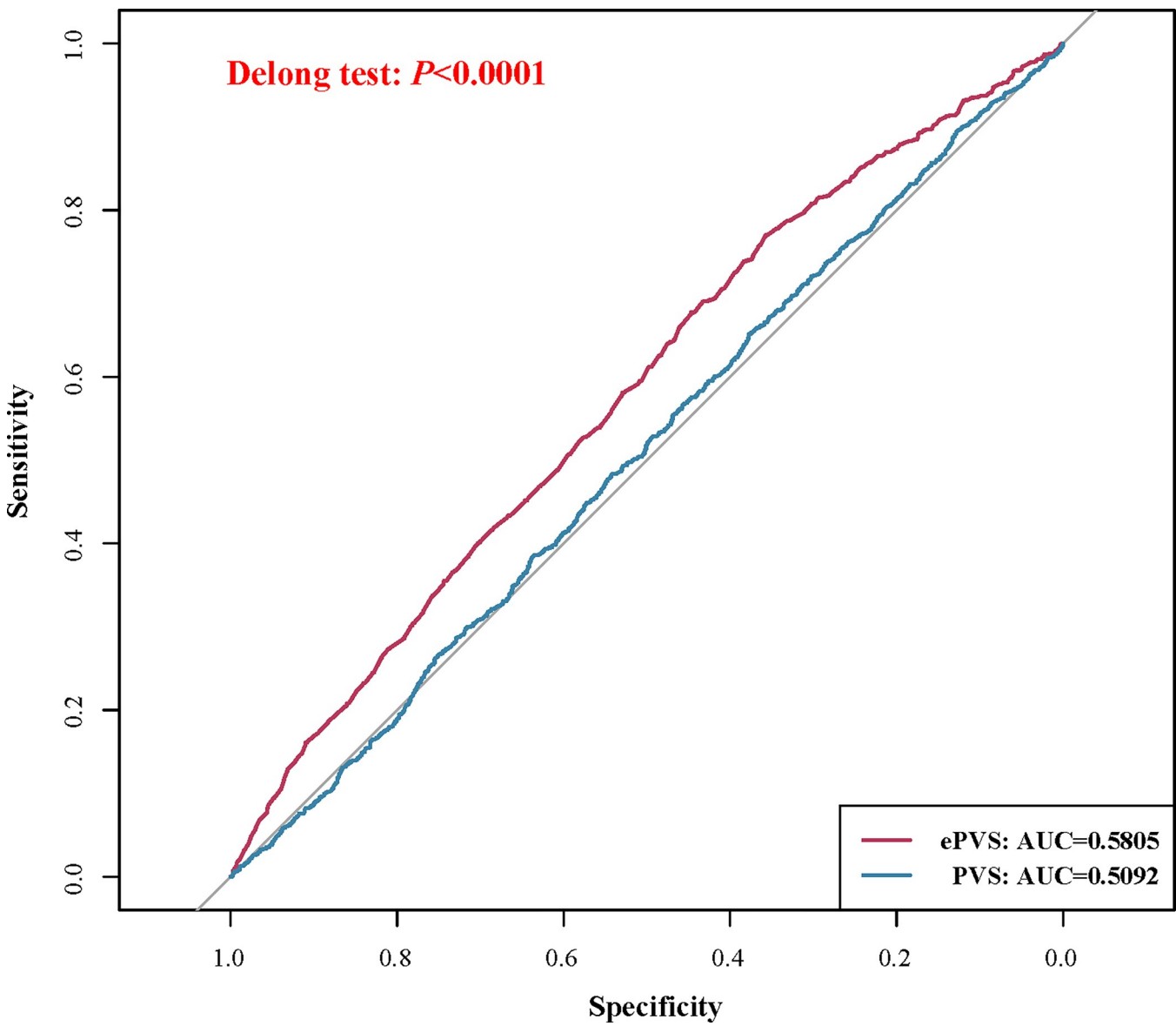

**Fig 2. The comparison of area under ROC curves for ePVS and PVS.**

AKI in patients undergoing PCI, which may be due to the more invasive procedure of PCI. Evidence shows that PCI, as compared with CABG, increases the rate of the combined end point of major adverse cardiac or cerebrovascular events at 1 year [32]. Due to the small sample size of these subgroups, there may be a population bias, and large samples of specific populations are needed to validate these results in the future.

Several possible mechanisms may explain the association between high ePVS and higher odds of AKI in patients who underwent coronary revascularization. Patients with higher levels of ePVS may be accompanied by progressive venous congestion, which further drives overactivation of the renin-angiotensin-aldosterone system leading to perpetuate water and sodium retention, and consequently promoting the development of AKI [33, 34]. Progressive ventricular dilatation leads to increased end-diastolic pressure and wall stress, which may exacerbate ischemia and cardiomyocyte loss, thereby contributing to advanced HF after CR [35]. Venous

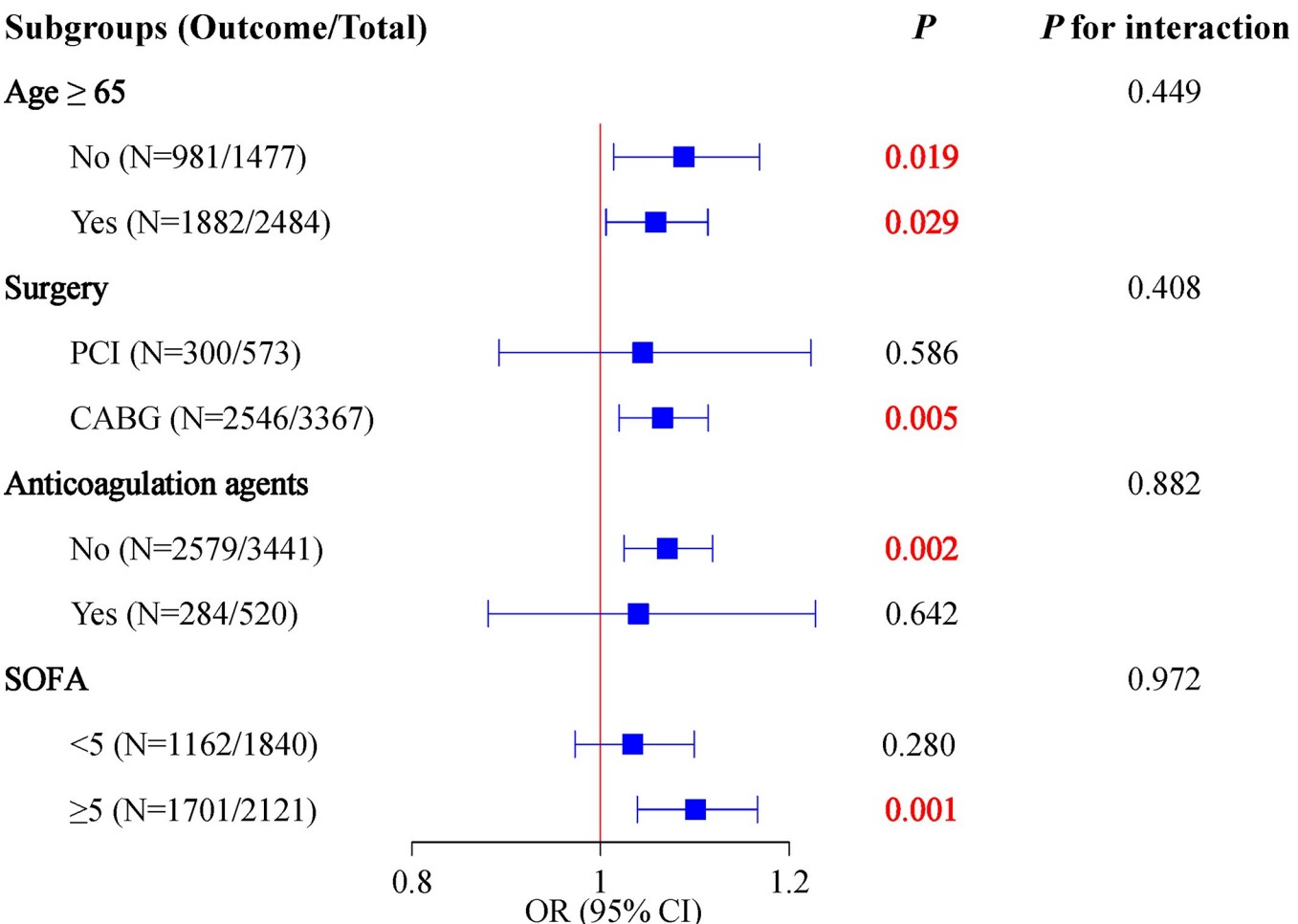

**Fig 3. Association between ePVS and the odds of AKI in patients who underwent coronary revascularization in subgroups of age, surgery, and anticoagulation agents.**

congestion can cause increased renal venous pressure, increase renal interstitial pressure, thereby impairs renal blood flow, destroys renal tissue structure, and then affects local oxygen exchange, leading to the propensity to renal injury seen after in CAD patients who underwent coronary revascularization [36–38].

The present study found an association between high ePVS was associated with the higher odds of AKI in patients who received coronary revascularization. Previously, PVS was usually measured by tracer dilution techniques, and in the context of old age, high comorbidity burden, and economic problems, this invasive outpatient monitoring may not be widely available to all coronary artery diseases patients. The ePVS based on changes in serial hemoglobin/hematocrit measurements could hence represent a low-cost and easy to use alternative, which may be a potential prognostic indicator for the pre-bed management of patients who received coronary revascularization. Monitoring the ePVS level in patients who underwent coronary revascularization can help clinicians early identify patients with high risk of AKI and provide certain reference for risk stratification management and early intervention treatment of these patients.

This study has several limitations, and caution should be exercised in interpreting our findings. First, this was a single-center retrospective cohort study, which might be selection bias

and reporting bias. Second, due to database limitations, brain natriuretic peptide (BNP), ejection fraction and the amount of contrast agent used during coronary revascularization was not recorded, which may have a certain impact on the results. Third, only ICU patients were explored in this study, and the application value of ePVS in routine hospitalized patients needs to be further study.

## Conclusion

The present study evaluated the associations of ePVS with the odds of AKI in patients who underwent coronary revascularization. We found the high ePVS was associated with higher odds of AKI in patients underwent coronary revascularization. Similar results were discovered in patients who underwent the CABG, without use anticoagulation agents use and with high SOFA score. The results might provide a reference for risk stratification and management of patients who underwent coronary revascularization.

## Supporting information

**S1 Fig. The process of Lasso regression screening covariates.** A: Path plot of Lasso coefficient; B: Lasso cross-validation plot.
(TIF)

**S1 Dataset.**
(CSV)

**S1 Table. Sensitivity analyses before and after imputation of missing data.**
(DOCX)

**S2 Table. The collinearity tests between covariates.**
(DOCX)

**S3 Table. Comorbidities of CCI in AKI and non-AKI.**
(DOCX)

**S4 Table. The ePVS level in different subgroups.**
(DOCX)

## Acknowledgments

The authors appreciate the time and effort of all participants. Also, the author would like to thank The MIMIC-IV database (MIMIC-IV v2.2 (physionet.org) for data support.

## Author Contributions

**Conceptualization:** Xinping Yang, Jiahua Shi.

**Data curation:** Fan Zhang, Yongqiang Zhan, Zhiheng Liu, Wenjing Wang.

**Formal analysis:** Fan Zhang, Yongqiang Zhan, Zhiheng Liu, Wenjing Wang.

**Investigation:** Fan Zhang, Yongqiang Zhan, Zhiheng Liu, Wenjing Wang.

**Methodology:** Fan Zhang, Yongqiang Zhan, Zhiheng Liu, Wenjing Wang.

**Writing – original draft:** Xinping Yang.

**Writing – review & editing:** Xinping Yang, Jiahua Shi.

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
