## [Decision Letter · Decision Letter 0]

14 Dec 2023

PONE-D-23-39649Association between estimated plasma volume status and acute kidney injury in patients who underwent coronary revascularization: a retrospective cohort study from the MIMIC-IV databasePLOS ONE

Dear Dr. Shi,

Thank you for submitting your manuscript to PLOS ONE. After careful consideration, we feel that it has merit but does not fully meet PLOS ONE’s publication criteria as it currently stands. Therefore, we invite you to submit a revised version of the manuscript that addresses the points raised during the review process.

We look forward to receiving your revised manuscript.

Kind regards,

Seunghwa Lee

Academic Editor

PLOS ONE

Journal Requirements:

Please ensure that your manuscript meets PLOS ONE's style requirements, including those for file naming. The PLOS ONE style templates can be found athttps://journals.plos.org/plosone/s/file?id=wjVg/PLOSOne_formatting_sample_main_body.pdf and
https://journals.plos.org/plosone/s/file?id=ba62/PLOSOne_formatting_sample_title_authors_affiliations.pdf

PLOS requires an ORCID iD for the corresponding author in Editorial Manager on papers submitted after December 6th, 2016. Please ensure that you have an ORCID iD and that it is validated in Editorial Manager. To do this, go to ‘Update my Information’ (in the upper left-hand corner of the main menu), and click on the Fetch/Validate link next to the ORCID field. This will take you to the ORCID site and allow you to create a new iD or authenticate a pre-existing iD in Editorial Manager. Please see the following video for instructions on linking an ORCID iD to your Editorial Manager account: https://www.youtube.com/watch?v=_xcclfuvtxQ

Please include a caption for figure 1.

Reviewers' comments:

Reviewer's Responses to Questions

**Comments to the Author**

1. Is the manuscript technically sound, and do the data support the conclusions?

Reviewer #1: Yes

Reviewer #2: Yes

Reviewer #3: Yes

2. Has the statistical analysis been performed appropriately and rigorously? 

Reviewer #1: Yes

Reviewer #2: Yes

Reviewer #3: N/A

3. Have the authors made all data underlying the findings in their manuscript fully available?

Reviewer #1: Yes

Reviewer #2: Yes

Reviewer #3: No

4. Is the manuscript presented in an intelligible fashion and written in standard English?

Reviewer #1: Yes

Reviewer #2: No

Reviewer #3: No

5. Review Comments to the Author

Reviewer #1: Thank you for the opportunity to review this manuscript. This was a retrospective cohort study using MIMIC-4 database. The authors demonstrated the association between estimated plasma volume status (ePVS) and the risk of AKI in patients undergoing coronary revascularization, which provides a promising parameter to evaluate the risk stratification management of ICU patients who underwent coronary revascularization. The study was generally well designed; However, there are a few additional points that need to be checked.

Major concern

1.As potential confounders, patients with anemia or polycythemia in the eligibility criteria are excluded or recognized.

2.Multi-factor regression must be comprehensively addressed. Does the equation indeed only consider the ePVS in its final form?

3.When conducting the regression analysis, it is important to consider collinearity, as there are multiple variables involved in the analysis.

4.The t-test and chi-square test value are not commonly used in statistical presentation. Some measures in the table are presented as medians, while others are presented as means?

5.The basis for sub-group grouping can be represented using a forest map.

6.The discussion section should be revised to enhance its clinical relevance.

Reviewer #2: Dear Author, thank you for the interesting article.

My clarifications are as follows:

1. Do clarify if the formula used to estimate the ePVS is the only formula available. Are there formulas that correct for gender differences and/or normalise for weight and height. If so, why was the formula in the study chosen as 1-Hct/Hb?

2. I have concerns that hematocrit and hemoglobin can be acutely affected by many issues that may not be directly linked to volume status. Is there data on proportion of patients who have hematological conditions, erythropoietin stimulating agents that can affect your ePVS calculation?

3. The incidence of AKI post coronary revascularisation can also be related to other factors such as overall cardiac function. A patient with poorer cardiac function such as low EF maybe at higher risk of AKI (part of cardio-renal syndrome) and correspondingly more likely to be overloaded and so have higher risk of AKI but I noted that there is no data on the EF in the data set. Also the amount of contrast given during coronary revascularisation is not known. Are these data available to be included

4. Is there any interaction between the ePVS and other dependent covariates that is strongly associated with AKI such as age and high SOFA scores?

5. Minor grammatical error - page 5 line 70 , "to the best" and not beast.

Many thanks

Reviewer #3: 1. Please explain when the patient received fluid resuscitation (plasma volume status were estimated) and when AKI was diagnosed?

2. The author(s) should discuss why the high ePVS was associated with a risk of AKI in patients aged ＜65 years, but not in patients aged ≥65?

3. Is there any differences in ePVS between patients aged ＜65 and ≥ 65 years old?

4. What were the comorbidities in patients with and without AKI respectively?

5. The English needs to be improved.

6. PLOS authors have the option to publish the peer review history of their article (what does this mean?). If published, this will include your full peer review and any attached files.

Reviewer #1: **Yes: **Ming Wu

Reviewer #2: No

Reviewer #3: **Yes: **Feng SHEN

---

## [Author Response · Author response to Decision Letter 0]

19 Feb 2024

Dear Seunghwa Lee

Thanks a lot for your letter and comments on our manuscript. We have revised the manuscript and given point-by-point responses according to the editor and reviewer comments, which have been highlighted in red. We hope that the revised manuscript is now acceptable for publication in your journal. 

Thanks again for what you have done for us and looking forward to your reply soon.

Best regards,

Xinping Yang

Reviewers' comments:

Reviewer's Responses to Questions

Comments to the Author: 

1. Is the manuscript technically sound, and do the data support the conclusions?

Reviewer #1: Yes

Reviewer #2: Yes

Reviewer #3: Yes

Reply: Thanks a lot for your comments.

2. Has the statistical analysis been performed appropriately and rigorously？

Reviewer #1: Yes

Reviewer #2: Yes

Reviewer #3: N/A

Reply: Thanks a lot for your comments.

3. Have the authors made all data underlying the findings in their manuscript fully available? 

Reviewer #1: Yes

Reviewer #2: Yes

Reviewer #3: No

Reply: Thanks a lot for your comments. We have presented all our results in the revised manuscript or in the supplement tables. 

4. Is the manuscript presented in an intelligible fashion and written in standard English? 

Reviewer #1: Yes

Reviewer #2: No

Reviewer #3: No

Reply: Thanks a lot for your comments. We have carried out a proofreading of the full manuscript.

Reviewer #1: Thank you for the opportunity to review this manuscript. This was a retrospective cohort study using MIMIC-4 database. The authors demonstrated the association between estimated plasma volume status (ePVS) and the risk of AKI in patients undergoing coronary revascularization, which provides a promising parameter to evaluate the risk stratification management of ICU patients who underwent coronary revascularization. The study was generally well designed; However, there are a few additional points that need to be checked.

Major concern

1.As potential confounders, patients with anemia or polycythemia in the eligibility criteria are excluded or recognized. 

Reply: Thanks a lot for your careful review of our manuscript. Data about anemia or polycythemia of study population in the MIMIC-IV were extracted. Finally, we discovered that fewer patients had polycythemia, so we included polycythemia as one of the exclusion population criteria and patients with polycythemia were excluded from this study. Given the large number of patients with anemia, anemia was considered as a confounding variable and used in subsequent covariate screening. 

The sample size of the study was changed due to the changes of the inclusion and exclusion criteria. Here we present the final study population screening flowchart and related results. 

2.Multi-factor regression must be comprehensively addressed. Does the equation indeed only consider the ePVS in its final form?

Reply: Thanks a lot for your comments. Plasma volume state (PVS) and ePVS were considered in our revised study. The PVS was calculated based on the patient’s body weight and hematocrit. Then, a percentage between the ideal plasma volume (iPV) and PV can be derived, and the difference in this percentage was referred to as the PVS. In the revised manuscript, we also used PVS as an exposure factor in our study to explore its association with the risk of AKI among patients after coronary revascularization. The results showed that there was no relationship between PVS and the risk of AKI among patients after coronary revascularization. Compared to PVS, ePVS can more accurately predict the risk of AKI among patients after coronary revascularization. 

3.When conducting the regression analysis, it is important to consider collinearity, as there are multiple variables involved in the analysis.

Reply: Thanks a lot for your comments. We evaluated the collinearity between individual covariates, and we performed collinearity tests for included covariates. The variance inflation factor (VIF) values of each variable were less than 10, indicating that there was no high collinearity between these variables. Supplement table 2 presents the results for collinearity. 

• 

4.The t-test and chi-square test value are not commonly used in statistical presentation. Some measures in the table are presented as medians, while others are presented as means? 

Reply: Thanks a lot for your comments. After carefully reviewing our manuscript, we took your advice and removed the statistic from the Table 1 of characteristics of the study population. Continuous variables with normal distribution were described as mean ± standard deviation (SD) and categorical variables were represented as number and percentage. 

• 

5.The basis for sub-group grouping can be represented using a forest map.

Reply: Thanks a lot for your comments. The sub-group grouping was depicted in the form of forest map in Figure 3. 

• 

6.The discussion section should be revised to enhance its clinical relevance. 

Reply: Thanks a lot for your comments. We supplemented the clinical significance of ePVS in predicting the risk of AKI among patients after coronary revascularization. Previously, plasma volume was usually measured by tracer dilution techniques, and in the context of old age, high comorbidity burden, and economic problems, this invasive outpatient monitoring may not be widely available to all coronary artery diseases patients. The ePVS based on changes in serial hemoglobin/hematocrit measurements could represent a low-cost and easy to use alternative, which may be a potential prognostic indicator for the pre-bed management of patients who received coronary revascularization. Monitoring the ePVS level in patients who underwent coronary revascularization can help clinicians early identify patients with high risk of AKI and provide certain reference for risk stratification management and early intervention treatment of these patients.

Reviewer #2: Dear Author, thank you for the interesting article.

My clarifications are as follows:

1. Do clarify if the formula used to estimate the ePVS is the only formula available. Are there formulas that correct for gender differences and/or normalise for weight and height. If so, why was the formula in the study chosen as 1-Hct/Hb?

Reply: Thanks a lot for your comments. The PVS and ePVS were considered in our study. The PVS was calculated based on the patient’s body weight and hematocrit. The association between PBS and the risk of AKI in patients who underwent the coronary revascularization was evaluated, and the results showed no statistical association between PVS and the risk of AKI in patients who underwent the coronary revascularization. The ePVS was calculated by Duarte et al. to estimate the PV derived from Strauss’s formula using plasma volume measurements derived “instantaneous” as a single time point [1]. However, clinically, PVS is usually measured by tracer dilution techniques, but these methods are expensive, time-consuming and complicated. Therefore, ePVS based on hemoglobin and hematocrit calculations have been proposed and confirmed by Duarte et al. [1] as an alternative to radioisotope tracer detection. Although there was no statistical association between PVS and the risk of AKI in patients who underwent the coronary revascularization in our study population, future large-scale studies are needed to confirm the role of PVS as a clinically important indicator of poor prognosis in coronary artery disease.

1. Duarte, K., et al., Prognostic Value of Estimated Plasma Volume in Heart Failure. JACC Heart Fail, 2015. 3(11): p. 886-93.

2. I have concerns that hematocrit and hemoglobin can be acutely affected by many issues that may not be directly linked to volume status. Is there data on proportion of patients who have hematological conditions, erythropoietin stimulating agents that can affect your ePVS calculation?

Reply: Thanks a lot for your comments. Data of the patients with blood diseases based on ICD-9 codes with 280-289 and ICD-10 codes with D50-D89 were extracted from the MIMIC-IV database and presented in Table 1. Among 3,961 patients, 2,083 (52.59%) had blood diseases. Due to the limitation of the MIMIC-IV database, there was no contain data on the treatment of erythropoietin stimulating agents. Given that the iron uses may affect the results of ePVS calculation, we replace the erythropoietin stimulating agents with iron use, also presented in Table 1. 

3. The incidence of AKI post coronary revascularization can also be related to other factors such as overall cardiac function. A patient with poorer cardiac function such as low EF maybe at higher risk of AKI (part of cardio-renal syndrome) and correspondingly more likely to be overloaded and so have higher risk of AKI but I noted that there is no data on the EF in the data set. Also the amount of contrast given during coronary revascularization is not known. Are these data available to be included?

Reply: Thanks a lot for your comments. Unfortunately, there were no data on EF in MIMIC-IV database, and only one patient had the amount of contrast. In view of the potential impact of these two factors on the risk of AKI after coronary revascularization, we consider them as the limitation of our study. In the future, larger cohort studies incorporating EF and the amount of contrast are needed to explore the association between ePVS and the risk of AKI in patients who underwent the coronary revascularization. 

4. Is there any interaction between the ePVS and other dependent covariates that is strongly associated with AKI such as age and high SOFA scores?

Reply: Thank you for your comments. The interaction between ePVS and age and high SOFA scores were evaluated. As depicted in Table S4, the results showed that there was no interaction between ePVS and age and SOFA. 

5. Minor grammatical error - page 5 line 70 , "to the best" and not beast.

Reply: Thank you for your careful review of our manuscript. We apologize for this error and have corrected it in the manuscript. 

Reviewer #3: 1. Please explain when the patient received fluid resuscitation (plasma volume status were estimated) and when AKI was diagnosed?

Reply: Thank you for your careful review of our manuscript. Plasma volume status was estimated by measurement of hematocrit and hemoglobin within 24 hours after ICU admission. AKI was diagnosed during the patients admitted in the ICU. The median follow-up time was 0.86 (0.60, 1.25) days. 

2. The author(s) should discuss why the high ePVS was associated with a risk of AKI in patients aged <65 years, but not in patients aged ≥65?

Reply: Thank you for your comments. The inclusion and exclusion criteria were revised according to the recommendations of the aforementioned reviewers. We reexamined the odds of ePVS and AKI among patients who underwent coronary revascularization based on age, surgery, anticoagulation agents and SOFA score. Finally, the results showed that regardless of whether the patients aged ≥65 years were all associated with a higher odds of AKI in patients who underwent coronary revascularization. Our study suggested that ePVS may have the ability to predict the odds of AKI in patients who underwent the coronary revascularization. 

3. Is there any differences in ePVS between patients aged <65 and ≥ 65 years old?

Reply: Thank you for your comments. We compared the ePVS between patients aged <65 years old and ≥65 years old and found that the ePVS level was different between the two age groups (P<0.001). The results were presented in Table S4. 

4. What were the comorbidities in patients with and without AKI respectively?

Reply: Thanks a lot for your comments. We explored the comorbidities in patients with and without AKI and presented the results in supplement table 3. 

5. The English needs to be improved. 

Reply: Thanks a lot for your comments. We have carried out a proofreading of the full manuscript.

6. PLOS authors have the option to publish the peer review history of their article (what does this mean?). If published, this will include your full peer review and any attached files.

Do you want your identity to be public for this peer review? For information about this choice, including consent withdrawal, please see our Privacy Policy.

Reviewer #1: Yes: Ming Wu

Reviewer #2: No

Reviewer #3: Yes: Feng SHEN

---

## [Decision Letter · Decision Letter 1]

4 Mar 2024

Association between estimated plasma volume status and acute kidney injury in patients who underwent coronary revascularization: a retrospective cohort study from the MIMIC-IV database

PONE-D-23-39649R1

Dear Dr. Shi,

We’re pleased to inform you that your manuscript has been judged scientifically suitable for publication and will be formally accepted for publication once it meets all outstanding technical requirements.

Kind regards,

Seunghwa Lee

Academic Editor

PLOS ONE

Additional Editor Comments (optional):

Reviewers' comments:

Reviewer's Responses to Questions

**Comments to the Author**

1. If the authors have adequately addressed your comments raised in a previous round of review and you feel that this manuscript is now acceptable for publication, you may indicate that here to bypass the “Comments to the Author” section, enter your conflict of interest statement in the “Confidential to Editor” section, and submit your "Accept" recommendation.

Reviewer #1: All comments have been addressed

Reviewer #2: All comments have been addressed

Reviewer #3: All comments have been addressed

2. Is the manuscript technically sound, and do the data support the conclusions?

Reviewer #1: Yes

Reviewer #2: Yes

Reviewer #3: Yes

3. Has the statistical analysis been performed appropriately and rigorously? 

Reviewer #1: Yes

Reviewer #2: Yes

Reviewer #3: Yes

4. Have the authors made all data underlying the findings in their manuscript fully available?

Reviewer #1: Yes

Reviewer #2: Yes

Reviewer #3: Yes

5. Is the manuscript presented in an intelligible fashion and written in standard English?

Reviewer #1: Yes

Reviewer #2: Yes

Reviewer #3: (No Response)

6. Review Comments to the Author

Reviewer #1: The manuscripts wihch we concerned have been revised. Thank you for the opportunity to review this manuscript again.

Reviewer #2: DO consider adding the limitations such as lack of EF/amount of contrast used as potential limitation to the discussion

Reviewer #3: (No Response)

7. PLOS authors have the option to publish the peer review history of their article (what does this mean?). If published, this will include your full peer review and any attached files.

Reviewer #1: No

Reviewer #2: No

Reviewer #3: **Yes: **Feng SHEN

---

## [Editor Report · Acceptance letter]

8 May 2024

PONE-D-23-39649R1 

PLOS ONE

Dear Dr. Shi, 

I'm pleased to inform you that your manuscript has been deemed suitable for publication in PLOS ONE. Congratulations! Your manuscript is now being handed over to our production team.

Kind regards, 

on behalf of

Dr. Seunghwa Lee 

Academic Editor

PLOS ONE